# META-PRIOR: META LEARNING FOR ADAPTIVE INVERSE PROBLEM SOLVERS

## ABSTRACT

Deep neural networks have become a foundational tool for addressing imaging inverse problems. They are typically trained for a specific task, with a supervised loss to learn a mapping from the observations to the image to recover. However, real-world imaging challenges often lack ground truth data, rendering traditional supervised approaches ineffective. Moreover, for each new imaging task, a new model needs to be trained from scratch, wasting time and resources. To overcome these limitations, we introduce a novel approach based on meta-learning. Our method trains a meta-model on a diverse set of imaging tasks that allows the model to be efficiently fine-tuned for specific tasks with few fine-tuning steps. We show that the proposed method extends to the unsupervised setting, where no ground truth data is available. In its bilevel formulation, the outer level uses a supervised loss, that evaluates how well the fine-tuned model performs, while the inner loss can be either supervised or unsupervised, relying only on the measurement operator. This allows the meta-model to leverage a few ground truth samples for each task while being able to generalize to new imaging tasks. We show that in simple settings, this approach recovers the Bayes optimal estimator, illustrating the soundness of our approach. We also demonstrate our method's effectiveness on various tasks, including image processing and magnetic resonance imaging.

## 1 INTRODUCTION

Linear inverse imaging problems consist of recovering an image through incomplete, degraded measurements. Typical examples include image restoration (Zhou et al., 2020; Liang et al., 2021), computed tomography (Bubba et al., 2019), magnetic resonance imaging (MRI; Knoll et al. 2020) and radio-astronomical imaging (Onose et al., 2016). While traditional techniques are based on variational approaches (Vogel, 2002), neural networks have progressively imposed themselves as a cornerstone to solve inverse imaging problems (Gilton et al., 2021; Genzel et al., 2022; Mukherjee et al., 2023). Given the knowledge of a measurement operator, and given a dataset of proxies for ground truth images, one can design a training set with input-target pairs for supervised learning (Zbontar et al., 2018). While such a strategy has proven tremendously efficient on some problems, such as image restoration and MRI, they remain difficult to use in many situations. As they require large training sets, their training is not possible when no proxy for ground truth data is available , for instance when no fully sampled data is available (*e.g.* in MRI, see Shimron et al. 2022), or when data is extremely scarce. Moreover, as each network is trained for a specific measurement operator, it does not generalize to other measurement operators and needs to be retrained when the operator changes.

Several techniques have been proposed to circumvent these drawbacks. To make the network adaptive to the operator, unrolled neural networks directly leverage the knowledge of the measurement operator in their architecture (Adler & Öktem, 2018; Hammernik et al., 2023) but these approaches remain sensitive to distribution shifts in both measurement operators and image distribution. Other adaptive approaches are plug-and-play methods (Venkatakrishnan et al., 2013; Romano et al., 2017; Ryu et al., 2019; Zhang et al., 2021a) and their recent diffusion extensions (Zhu et al., 2023), which use generic denoiser to inject prior information into variational solvers. These methods are adaptive to the operator as the variational backbone accounts for the sample-specific measurement operator. Yet, the performance of the associated implicit prior tends to decrease when applied to data beyond its training distribution, limiting its ability to generalize. Moreover, the chosen architecture and training must be constrained to ensure the stability and convergence of the variational method (Pes-

quet et al., 2021; Hurault et al., 2021). When few to no examples are available, self-supervised training losses have been proposed leveraging equivariant properties of both the target data distribution and the measurement operators (Chen et al., 2022). Data augmentation also has a significant impact on the robustness of the model in this setting and its ability to generalize to unseen distribution (Rommel et al., 2021).

Meta-learning provides a framework for enabling efficient generalization of trained models to unseen tasks (Finn et al., 2017; Raghu et al., 2020; Rajeswaran et al., 2019; Hospedales et al., 2021). Instead of training a model on a fixed, single task similar to the one that will be seen at test time, a so-called meta-model is trained simultaneously on multiple tasks, while ensuring that its state is close to the optimal state for each individual task. As a consequence, the resulting meta-model can be seen as a barycenter of optimal states for different tasks and appears as an efficient initialization state for fine-tuning the meta-model on new tasks. This approach has proven successful in a variety of domains, prominent examples including few-shot learning, reinforcement learning for robotics, and neural architecture search, among others (Alet et al., 2018; Elsken et al., 2020).

In this work, we present a meta-learning strategy that involves training a versatile meta-model across a range of imaging tasks while fine-tuning its inner states for task-specific adaptation. We explore the bilevel formulation of the meta-learning problem, leading to novel self-supervised fine-tuning approaches, particularly valuable in scenarios where ground truth data is unavailable. More precisely, we show that the meta-model can be fine-tuned on a specific task with a loss enforcing fidelity to the measurements only, without resorting to additional assumptions on the target signal. We analyze the dynamics of the learned parameters, showing that task-specific parameters adapt to the measurement operator, while the meta-prior completes the missing information in its kernel. Our experiments provide empirical evidence of the approach's effectiveness, demonstrating its ability to fine-tune models in a self-supervised manner for previously unseen tasks. The proposed approach also demonstrates good performance in a supervised setup. In a broader context, our findings suggest that meta-learning has the potential to offer powerful tools for solving inverse problems.

## 2 RELATED WORKS

The meta-learning framework has seen extensive use in various computer vision tasks but has yet to be fully explored in the context of solving inverse problems in imaging. Nevertheless, its underlying bilevel optimization formulation shares similarities with techniques commonly employed in addressing imaging inverse problems, e.g. for hyperparameter tuning. In this section, we offer a concise literature review, shedding light on the potential synergies between meta-learning and well-established methods for tackling challenges in imaging.

**Meta learning for vision tasks** Due to its successes in task adaptation to low data regimes, meta-learning is widely spread in various fields of computer vision, proposing an alternative to other widely used self-supervised techniques in the computer vision literature (Chuang et al., 2020; Caron et al., 2021; Dufumier et al., 2023; Walmer et al., 2023). It has demonstrated successes on a few shot learning tasks, where one aims at fine-tuning a model on a very limited amount of labeled data; for instance in image classification (Vinyals et al., 2016; Khodadadeh et al., 2019; Chen et al., 2021b;a), in image segmentation (Tian et al., 2020; Yang et al., 2020) or in object detection (Wang et al., 2019; Zhang et al., 2022). Despite recent links between meta-learning and image-to-image translation (Eißler et al., 2023), its utilization in such tasks remains relatively uncommon.

**Unsupervised meta-learning** In Khodadadeh et al. (2019), the authors propose a method for unsupervised meta-learning, relying on augmentation techniques. This is reminiscent of methods such as equivariant learning or constrastive learning. Antoniou & Storkey (2019) propose to mix unsupervised and supervised loss terms at the inner problem, akin to what we propose in this work. We note that in general, efforts to improve the generalization capabilities of meta-learning (or related approaches) often rely on strategies leveraging both the discrete nature of the classification problem and the encoding nature of the problem that are specific to classification tasks (Snell et al., 2017; Zhang et al., 2022), which do not apply to image-to-image translation tasks. For instance, the meta learning from Xu et al. (2021) relies on both cluster embedding and data augmentation.

**Model robustness in imaging inverse problems** Deep neural networks for imaging inverse problems, and more generally, for image-to-image translation tasks, tend to be trained in a supervised

fashion on datasets containing the operators that will be seen at test time. This is the case in MRI imaging (Zbontar et al., 2018; Knoll et al., 2020) or in image restoration (Zhang et al., 2021b; Zhou et al., 2020). The efficiency and robustness of the model then strongly rely on the diversity of the training set, thus sparking interest in augmenting the dataset with realistic samples (Rommel et al., 2021; Zhang et al., 2021b). In order to lighten the dependency on the measurement operator, Chen et al. (2022) show that the neural network can be trained without resorting to the use of ground truth data, solely relying on the equivarience properties of the measurement operator. Their method results in a fully unsupervised training setting.

**Bilevel optimization for imaging tasks** A longstanding problem in using variational methods for solving inverse problems is the choice of hyper-parameters; bilevel optimization techniques have been proposed to fine-tune these parameters efficiently (Kunisch & Pock, 2013; Ochs et al., 2015; Holler et al., 2018). The recent work Ghosh et al. (2022) proposes to learn a convex model tool similar to those used in the implicit meta-learning literature. In Riccio et al. (2022), the authors propose a bilevel formulation of the training of a deep equilibrium model, where the inner problem computes the limit point of the unrolled model.

**Notations** Let $A$ be a linear operator; $A^\top$ denotes the adjoint of $A$ and $A^\dagger$ its Moore-Penrose inverse; $\mathrm{Ker}(A)$ and $\mathrm{Im}(A)$ denote the kernel and range of $A$, respectively. For a function $f : X \to Y$, we denote by $f\big|_S$ the restriction of $f$ to the set $S \subset X$.

## 3 META LEARNING FOR INVERSE PROBLEMS

Instead of learning a specific model for various different tasks, the idea of meta-learning is to learn a shared model for all the tasks, that can be easily adapted to each task with a few steps of fine-tuning (Finn et al., 2017). We propose to extend this approach to the context of inverse imaging problems. In this context, we consider that we have $I$ imaging tasks $\{\mathcal{T}_i\}_{i=1}^I$. Each of these tasks is described through a given linear operator $A_i \in \mathbb{R}^{m_i \times n}$ and a set of examples $\mathcal{D}_i = \{(x_j^{(i)}, y^{(i)})\}_{j=1}^{N_i}$, where $x_j^{(i)} \in \mathbb{R}^n$ is the image to recover and $y_j^{(i)} = A_i x_j^{(i)} + \epsilon_j^{(i)} \in \mathbb{R}^{m_i}$ is the associated measurement. Traditional approaches learn a model $f_{\theta_i}$ for each of the task $\mathcal{T}_i$ by minimizing either the supervised or the unsupervised loss:

$$
\theta_i = \underset{\theta}{\mathrm{argmin}} \ \mathcal{L}_{\mathrm{sup}}(f_\theta, \mathcal{T}_i, \mathcal{D}_i) = \sum_{j=1}^{N_i} \frac{1}{2} \big\| f_\theta(y_j^{(i)}, A_i) - x_j^{(i)} \big\|_2^2 \ ,
$$

$$
\text{or} \quad \theta_i = \underset{\theta}{\mathrm{argmin}} \ \mathcal{L}_{\mathrm{uns}}(f_\theta, \mathcal{T}_i, \mathcal{D}_i) = \sum_{j=1}^{N_i} \frac{1}{2} \big\| A_i f_\theta(y_j^{(i)}, A_i) - y_j^{(i)} \big\|_2^2 \ .
$$

(1)

While the supervised loss requires ground truth data $x_j^{(i)}$, the unsupervised loss only requires access to the measured data $y_j^{(i)}$. In both cases, the learned model $f_{\theta_i}$ cannot be used for other tasks $\mathcal{T}_k$ as it is not adaptive to the operator $A_k$.

The meta-learning strategy consists in training a model $f_\theta$ not only on one task but on a set of tasks $\mathcal{T}_i$ while ensuring that the model $f_\theta$ can be adapted to each task $\mathcal{T}_i$ with a few steps of fine-tuning. As proposed originally by Finn et al. (2017), this so-called meta-model is trained on various tasks simultaneously, while the fine-tuning is performed by a single step of gradient descent. In its implicit form (Rajeswaran et al., 2019), the meta-model solves the following bilevel optimization problem:

$$
\theta^* = \underset{\theta}{\mathrm{argmin}} \ \sum_{i=1}^I \mathcal{L}_{\mathrm{outer}}(f_{\theta_i}, \mathcal{T}_i, \mathcal{D}_i^{\mathrm{test}})
$$

$$
\text{s.t.} \ \theta_i = \underset{\phi}{\mathrm{argmin}} \ \mathcal{L}_{\mathrm{inner}}(f_\phi, \mathcal{T}_i, \mathcal{D}_i^{\mathrm{train}}) + \frac{\lambda}{2} \|\phi - \theta^*\|^2 , \quad \forall i \in \{1, \dots I\} \ .
$$

(2)

Here, $\mathcal{D}_i^{\mathrm{train}}$ and $\mathcal{D}_i^{\mathrm{test}}$ denote respectively the training and test datasets for the task $\mathcal{T}_i$, that are used to control that the model $f_{\theta_i}$ generalizes well to the task $\mathcal{T}_i$. The inner training loss $\mathcal{L}_{\mathrm{inner}}$ corresponds to the loss used to learn the model's parameters $\theta_i$ for a given task. It can be either the supervised loss $\mathcal{L}_{\mathrm{sup}}$ or the unsupervised loss $\mathcal{L}_{\mathrm{uns}}$, with an extra regularization term controlled by $\lambda > 0$ to

ensure that the model $f_{\theta_i}$ is close to the meta-model $f_{\theta^*}$. When $\mathcal{L}_{\text{inner}}$ uses $\mathcal{L}_{\text{sup}}$ (resp. $\mathcal{L}_{\text{uns}}$), we call this problem the *supervised meta-learning* (resp. *unsupervised meta-learning*). Finally, the outer training loss $\mathcal{L}_{\text{outer}}$ is used to evaluate how well the model $f_{\theta_i}$ generalizes on the task $\mathcal{T}_i$, using $\mathcal{L}_{\text{sup}}$.

Essentially, the interest of this bilevel formulation arises from the inner problem's resemblance to a fine-tuning procedure on a task $\mathcal{T}_i$ from the meta-model's state $\theta^*$. If the number of tasks $I$ presented during training is large enough, this model can be adapted to a novel unseen task $\mathcal{T}_k$ by solving the inner problem from (2) for the new task of interest on a small dataset. While both supervised and unsupervised formulations require access to some original data in the outer loss, it is important to note that in the unsupervised formulation, the fine-tuning to a new task can be performed without the need for ground truth data. This partially addresses the need for ground truth data to solve the inverse problem, as a model can be adapted to a given task can without accessing clean signals. Moreover, in this meta-learning framework, models aggregate information from the multiple inverse problems seen during training, as in both formulations, the weights of each partially fine-tuned model benefit from samples of other tasks through the regularization with the distance to $\theta^*$.

It is well known that a major challenge in unsupervised inverse problems is to correctly estimate the original data $x$ in $\text{Ker}(A_i)$ (Chen et al., 2022; Malézieux et al., 2023). Indeed, $\mathcal{L}_{\text{uns}}$ does not vary when the estimated solution moves in the kernel of the measurement operator. In the following, we investigate how meta-learning leverages multiple tasks to overcome this challenge.

**Learning meta-priors for linear models**  In order to analyze the meta-prior learned with our approach, we restrict ourselves to a linear model for a noiseless inverse problem. More precisely, given a linear operator $A_i$ and a signal $x$, we aim to estimate $x$ from measurements $y = A_i x$ using a linear model $f_\theta(y) = \theta y$. Our bilevel problem thus reads

$$\theta^* = \underset{\theta}{\text{argmin}} \sum_{i=1}^{I} \sum_{x,y \sim \mathcal{D}_i^{\text{test}}} \frac{1}{2}\|x - \theta_i y\|_2^2$$

$$\text{s.t. } \theta_i = \underset{\phi}{\text{argmin}} \sum_{x,y \sim \mathcal{D}_i^{\text{train}}} \frac{1}{2}\|A_i \phi y - y\|_2^2 + \frac{\lambda}{2}\|\phi - \theta\|^2 \tag{3}$$

where $\mathcal{D}_i^{\text{train}}$ and $\mathcal{D}_i^{\text{test}}$ are respectively the train and test datasets, containing $x$ and $y$ samples. The following result quantifies the behavior of $\theta^*$ and $\theta_i$ relative to the kernel of the measurement operator $A_i$.

**Theorem 3.1.** *Consider Problem* (3) *and assume that for all* $i$, $y_i \in \text{Im}(A_i)$. *Then*

(i) *During fine-tuning on a task* $\mathcal{T}_i$ *(in either supervised or unsupervised settings), the fine-tuned weight* $\theta_i$ *satisfies* $\Pi_{\text{Ker}(A_i)} \theta_i = \Pi_{\text{Ker}(A_i)} \theta^*$.

(ii) *Moreover, if the fine-tuning is performed with gradient descent and initialized at* $\theta^*$, *it holds at any step* $t \in \mathbb{N}$ *during optimization that the iterates* $(\theta_i^{(t)})_{t \in \mathbb{N}}$ *satisfy* $\Pi_{\text{Ker}(A_i)} \theta_i^{(t)} = \Pi_{\text{Ker}(A_i)} \theta^*$.

(iii) *Assume that* $\bigcap_i \text{Ker}(A_i) = \{0\}$ *and that* $\sum_j A_i x_i^{(j)} x_i^{(j)\top} A_i^\top$ *is full rank. Then the outer-loss for training the meta-model admits a unique minimizer.*

The proof is deferred to Section A.1. We can draw a few insightful observations from these results. First, (i) shows that the meta-prior plays a paramount role in selecting the right solution in $\text{Ker}(A_i)$. Indeed, as the gradient of the inner loss does not adapt the solution on the kernels of the $A_i$, its value is determined by the outcome of the meta-model on $\text{Ker}(A_i)$. This observation also holds for any number of steps used to solve the inner problem with gradient descent (ii), with an approach compatible with the standard MaML framework (Finn et al., 2017). Second, we notice from (iii) that as the number of tasks $I$ grows, the dimension of the nullspace restriction on the outer loss gradient decreases. In the limiting case where $\bigcap_i \text{Ker}(A_i) = \emptyset$, (iii) implies the existence of a unique solution to problem (3). This suggests that increasing the number of tasks improves the model's adaptability to novel, previously unseen tasks $\mathcal{T}_k$. As a side note, we notice that the image space of the hypergradient is not penalized by the unsupervised nature of the inner problems.

We stress that this approach differs from other unsupervised approaches from the literature, where one manages to learn without ground truth by relying on invariance properties, e.g. equivariance of

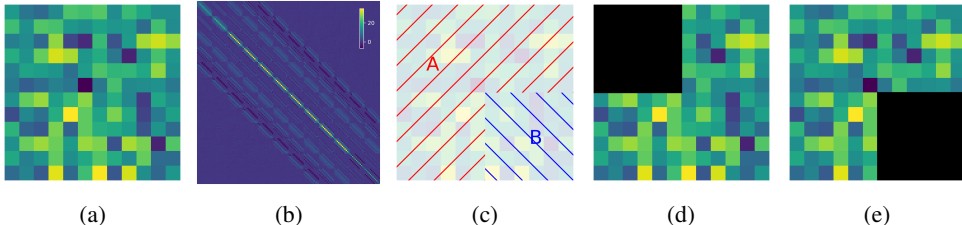

|  (a)  |  (b)  |  (c)  |  (d)  |  (e)  |

Figure 1: Illustration of our toy experimental setting: (a) a toy data sample; (b) covariance matrix from which (a) is sampled; (c) areas from which masks are sampled during train and test times; (d) the sample from (a) masked with a mask sampled from the training set; (e) the sample from (a) masked with a mask from the test set.

the measurements with respect to some group action (Chen et al., 2022). Here, we instead suggest taking advantage of multiple measurement operators and datasets to learn meta-models with features compensating for the lack of information and distribution bias inherent to each inverse problem and dataset, in line with ideas developed by Malézieux et al. (2023). Thus, our approach avoids relying on potentially restrictive equivariance assumptions on the signal or measurement operator. This is in particular illustrated by (i/ii): if the meta-model has not learned a good prior in the kernel of $A_i$, the fine-tuning cannot bring any improvement over the meta-model weights.

In order to demonstrate that this approach is adapted to learn interesting priors, we consider a simple task where the goal is to recover multivariate Gaussian data from degraded observations. More precisely, we assume that the samples $x$ are sampled from a Gaussian distribution $\mathcal{N}(\mu, \Sigma)$, and we show that the Bayes optimal estimator is related to the solution of Problem (3). Recall that the Bayes' estimator is defined as

$$\widehat{x}(y, A_i) = \underset{x'(y,A_i)\in\mathbb{R}^n}{\arg\min}\ \mathbb{E}\left[\|x - x'(y,A_i)\|_2^2\right] = \mathbb{E}[x|y, A_i], \tag{4}$$

where $y = A_i x$. Since $A_i$ is linear, one can derive a closed-form expression for the estimator $\widehat{x}$. We have the following result.

**Lemma 3.2.** *Let $A_i$ a linear operator and assume that $x \sim \mathcal{N}(\mu, \Sigma)$ and $y = A_i x$. Then the Bayes' estimator* (4) *satisfies:*

$$\begin{cases} \widehat{x}(y, A_i)_{\mathrm{Im}(A_i^\top)} = A_i^\dagger y\big|_{\mathrm{Im}(A_i^\top)}, \\ \widehat{x}(y, A_i)_{\mathrm{Ker}(A_i)} = \mu_{\mathrm{Ker}(A_i)} + \Sigma_{\mathrm{Ker}(A_i),\mathrm{Im}(A_i^\top)}\left(\Sigma_{\mathrm{Im}(A_i^\top)}\right)^{-1}(A_i^\dagger y - \mu_{\mathrm{Im}(A_i^\top)}), \end{cases} \tag{5}$$

*where we have used the decomposition*

$$\mu = \begin{pmatrix} \mu_{\mathrm{Im}(A_i^\top)} \\ \mu_{\mathrm{Ker}(A_i)} \end{pmatrix} \quad \text{and} \quad \Sigma = \begin{pmatrix} \Sigma_{\mathrm{Im}(A_i^\top)} & \Sigma_{\mathrm{Im}(A_i^\top),\mathrm{Ker}(A_i)} \\ \Sigma_{\mathrm{Ker}(A_i),\mathrm{Im}(A_i^\top)} & \Sigma_{\mathrm{Ker}(A_i)} \end{pmatrix}.$$

The proof is given in Section A.2. We stress that this general result goes beyond the meta-learning framework and can be applied in the general supervised learning setting with inverse problems' solutions. Indeed, Bayes' estimator can be seen as a solution to the (empirical risk) minimization problem at the outer level of (3), regardless of the inner level problem. Yet, this result complements Theorem 3.1 by giving an intuition on the expression for the model $\theta_i$ and the estimate $\widehat{x}(y, A_i) = \theta_i y$ that easily decompose on $\mathrm{Ker}(A_i)$ and $\mathrm{Im}(A_i^\top)$. We notice that on $\mathrm{Im}(A_i^\top)$, the solution is defined by the pseudoinverse $A_i^\dagger$, smoothed by the added regularization. On the kernel space of $A_i$, the distribution of the signal $x$ comes into play, and the reconstructed signal is obtained from $\theta^*$, as the weighted mean of terms of the form $\Sigma_{\mathrm{Ker}(A_i),\mathrm{Im}(A_i^\top)}\left(\Sigma_{\mathrm{Im}(A_i^\top)}\right)^{-1}A_i^\dagger$. This shows that meta-learning is able to learn informative priors when it can be predictive of the value of $\mathrm{Ker}(A_i)$ from the value of $\mathrm{Im}(A_i)$. In particular, we stress that in the case of uncorrelated signals $x$, *i.e.* when $\Sigma$ is diagonal, the second line of (5) becomes $\widehat{x}(y, A_i)_{\mathrm{Ker}(A_i)} = \mu_{\mathrm{Ker}(A_i)}$.

We now illustrate the practical generalization capabilities of the proposed approach in such a setting. We consider a setting where, during training, the operators $\{A_i\}_{i=1}^I$ are binary, square mask operators of fixed size, sampled at random from the red area denoted by A in Figure 1 (c). Similarly, the

| Learned weights (train) | Analytic solution (train) | Learned weights (test) | Analytic solution (test) |

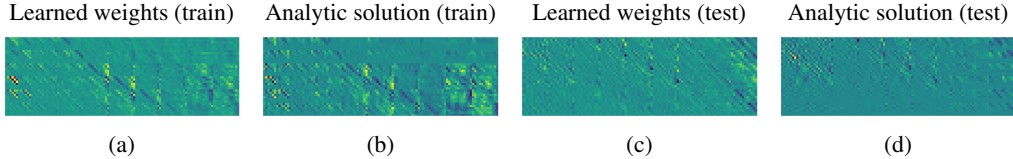

| (a) | (b) | (c) | (d) |

Figure 2: Learning $\widehat{x}(y, A_i)$ with a linear model for an inpainting task, in unsupervised and supervised settings. Each plot shows the matrix mapping between the observed data space $\mathrm{Im}(A_i)$ and $\mathrm{Ker}(A_i)$, the analytic solution being given by (5). (a) and (b) show learned weights and the analytic solution on training tasks. (c) and (d) show learned weights and the analytic solution on test tasks, with masks unseen during training and unsupervised fine-tuning loss.

test tasks consist of masking the bottom right corner (area denoted by B). We solve the inner problem approximately with 20 steps of Adam (Kingma & Ba, 2015) and the differentiation of the outer loss is computed via automatic differentiation. After the training phase, we fine-tuned the network with the unsupervised inner loss on the test task.

We show the weights learned by the network $\theta^*$ in Figure 2, on both a training task and after unsupervised fine-tuning on a test task. We see a strong similarity between the learned patterns and Bayes' optimal estimator on both the training tasks and the unsupervised fine-tuning on test tasks. Notice however that the weights learned on the training tasks match more closely to the analytical solution than the weights fine-tuned in an unsupervised fashion on the test task. We stress that in both cases, the learned solution does not match exactly the analytical solution, which we attribute to the stochastic nature of the learning procedure. Fine-tuning the model on a test task with masks on areas of the image that were not probed during training converges to weights that also show important similarity with Bayes' optimal estimator. This experiment confirms that the proposed approach allows learning a prior in the case of a linear model and that the influence of this prior percolates at the level of the fine-tuning, allowing the model to generalize to tasks unseen during training.

**Extention to the nonlinear case**    In the case of nonlinear models $f_\theta$, the influence of the nullspace of the $A_i$s is not as clear. We can however derive the following proposition in the case of unsupervised inner losses.

**Proposition 3.3.** *Consider Problem* (2). *If the network $f_\theta$ is extremely overparametrized and $\Pi_{\mathrm{Ker}(A_i)}$ commutes with $J_{\theta^*} J_{\theta^*}^\top$, then, we have*

$$f_\theta(y)\big|_{\mathrm{ker}(A_i)} = f_{\theta^*}(y)\big|_{\mathrm{ker}(A_i)} \ ,$$

*when the inner loss is unsupervised.*

This result shows that in the unsupervised setting, when the model $f_{\theta_i}$ has a simple mapping from its parameter space to the image space, it only adapts its output to the task in $\mathrm{Ker}(A_i)^\perp$. Intuitively, the model's inner loss remains 0 on $\mathrm{Ker}(A_i)$, and thus no gradient information is propagated. Under the assumption of Proposition 3.3, we recover the observation that was made in the linear case, *i.e.* that the supervision at the outer level of the meta-learning approach is necessary in order to capture meaningful information in the kernel of the $A_i$.

However, the commutativity assumption is very restrictive and unlikely to be satisfied in practice. Yet, we conjecture that in the highly overparametrized case, the result does hold. This result however suggests that the relation between the kernel of $A_i$ and the neural tangent kernel $J_\theta J_\theta^\top$ (NTK; Jacot et al. 2018) should be further explored in the light of Wang et al. (2022)'s work.

## 4    IMAGING EXPERIMENTS

In this section, we apply the proposed method to an unfolded architecture that is trained to solve (2) on different standard image restoration tasks in both supervised and unsupervised settings and investigate the generalization capabilities of the model to tasks unseen during training. While the previous sections focused on noiseless inverse problems only, we here consider more general problems during training, including image denoising and pseudo-segmentation.

### 4.1 PROBLEM FORMULATION

During training, we propose to solve Problem (2) for 4 different imaging tasks $\{\mathcal{T}_i\}_{i=1}^4$. The task $\mathcal{T}_1$ is image denoising, *i.e.* one wants to estimate $\overline{x}$ from $y = \overline{x} + \sigma e$ where $e$ is the realization of Gaussian random noise. $\mathcal{T}_2$ is total variation estimation (Condat, 2017), *i.e.* one wants to estimate $\mathrm{prox}_{\mathrm{TV}}(y)$ from $y$[1]. We underline that this problem is of interest since it can be seen as a simplified segmentation task (Chan et al., 2006). $\mathcal{T}_3$ is (noiseless) image deconvolution, *i.e.* one wants to estimate $y = k * \overline{x}$ for $k$ some convolutional kernel and $*$ the usual convolution operation. Eventually, $\mathcal{T}_4$ is image inpainting, *i.e.* one wants to estimate $\overline{x}$ from $y = M \odot x$ where $\odot$ denotes the elementwise (Hadamard) product and $M$ is a binary mask.

The selection of these training tasks serves a dual purpose: first, to foster the acquisition of emergent priors from a broad and varied training set, and second, to ensure a minimal overlap in the kernels of the measurement operators as our analysis suggests.

We propose to apply our model to two tasks that were not seen during training, namely image super-resolution and MRI. For natural images, we consider the Set3C test dataset as commonly used in image restoration tasks (Hurault et al., 2021); for MRI, we use a fully sampled slice data from the validation set of fastMRI (Zbontar et al., 2018).

### 4.2 ARCHITECTURE AND TRAINING

**PDNet architecture** Recent findings, as highlighted in (Yu et al., 2023), underscore the predominant role of architectural choice in the emergence of priors. In this work, we consider an unfolded Primal-Dual network (PDNet). PDNet takes its roots in aiming at solving the problem

$$\operatorname*{argmin}_x \frac{1}{2}\|Ax - y\|_2^2 + \lambda\|Wx\|_1. \tag{6}$$

This problem can be solved with a Primal-Dual algorithm (Chambolle & Pock, 2011), and each PDNet layer reads:

$$\begin{aligned} x_{k+1} &= x_k - \tau A^\top (A(x_k - y)) - \tau W_k^\top u_k \\ u_{k+1} &= \mathrm{prox}_{\gamma(\lambda_k\|\cdot\|_1)^*}(u + \gamma W_k(2x_{k+1} - x_k)), \end{aligned} \tag{7}$$

where $W$ is a linear operator (typically a sparsifying transform (Daubechies et al., 2004)) and $\lambda > 0$ a regularization parameter. One layer $f_{W_k,\lambda_k}$ of our network thus writes as the above iteration, and the full network writes:

$$f_{W,\lambda}(y) = f_{W_K,\lambda_K} \circ \cdots \circ f_{W_2,\lambda_2} \circ f_{W_1,\lambda_1}(y), \tag{8}$$

where $\tau, \gamma > 0$ are small enough stepsizes (Condat, 2013) and $(\cdot)^*$ denotes the convex conjugate. We stress that the weights and thresholding parameters vary at each layer; as a consequence, PDNet does not solve (6). It however offers a simple architecture taking into account the measurement operator $A_i$, improving the generalization capabilities of the network. Similar architectures are widely used in the imaging community (Adler & Öktem, 2018; Ramzi et al., 2020; 2022). More generally, this architecture has connections with several approaches and architectures from the literature: variational approaches when weights are tied between layers (Le et al., 2023), dictionary-learning based architectures (Malézieux et al., 2021) and residual architectures (Sander et al., 2022).

**Training details** For every $i$, the training dataset $\mathcal{D}_i^{\mathrm{train}}$ consists in the BSD500 training and test set combined (Arbelaez et al., 2011); the test dataset $\mathcal{D}_i^{\mathrm{test}}$ consists in the validation set of BSD500. The fine-tuning dataset depends on the task of interest and are detailed in Section 4.3. We train various versions of PDNet with depths $K$ ranging in $\{20, 30, \ldots, 120\}$. Each $(W_k)_{1 \le k \le K}$ is implemented as a convolutional filter with size $3 \times 3$, with 1 (resp. 40) input (resp. output) channels[2]. During training, we minimize the loss (2) where the training tasks $\{\mathcal{T}_i\}_{i=1}^4$ are described in Section 4.1. The solution to the inner problem in (2) is now approximated with only 1 step of Adam. This number is chosen to reduce the computational burden for deeper networks. We emphasize that the MaML

---

[1]For a convex lower-semicontinuous function $h$, the proximity operator of $h$ is defined as $\mathrm{prox}_h(y) = \operatorname*{argmin}_x h(x) + \frac{1}{2}\|x - y\|_2^2$.

[2]As a consequence, the PDNet architecture contains 361 learnable parameters per layer (7).

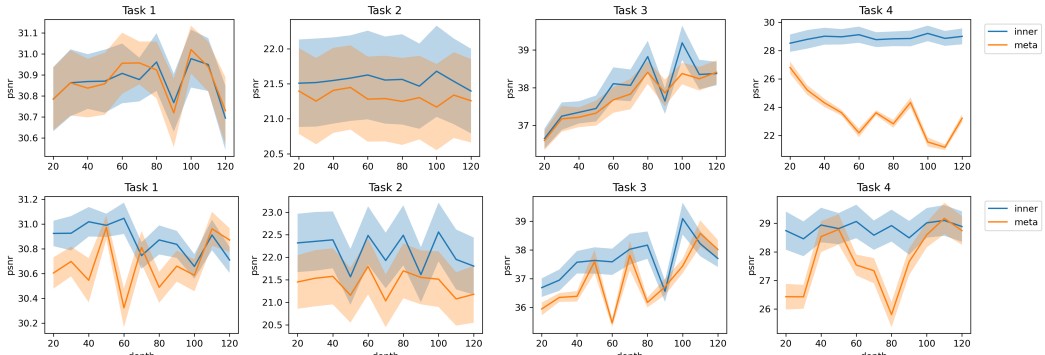

Figure 3: Mean reconstruction PSNR on the Set3C test set on tasks seen during training. The shaded area represents the empirical standard deviation. Top row: models trained with supervised inner and outer losses (supervised setting). Bottom row: models trained with supervised outer loss and unsupervised inner loss (unsupervised setting).

approach is memory intensive in the imaging context and efficient training optimization methods may be required when applying the proposed approach to larger models, such as UNets.

We display in Figure 3 the performance of the meta-model and task-specific models on the test set for tasks seen during training. In both supervised and unsupervised setups, inner models perform better than meta-models for all tasks, except for the denoising task (task 1) in the supervised setting. For the deblurring and inpainting tasks (tasks 3 and 4), the gap between the meta and inner model tends to increase with the depth of the model. Visual results are provided in Figure B.1. The performance of the inner models is similar in both supervised and unsupervised settings, with the supervised models achieving slightly better results.

### 4.3 GENERALIZATION TO UNSEEN TASKS

Next, we test our trained models on two tasks that were not seen during training, namely natural image super-resolution (SR) and MRI imaging. In this setting, we fine-tune the model as per the inner-problem formulation in (2), with either supervised or unsupervised losses. We compare the reconstructions obtained with the proposed methods with traditional handcrafted prior-based methods (namely wavelet and TGV (Bredies et al., 2010) priors) as well as with the state-of-the-art DPIR algorithm (Zhang et al., 2021a), which relies on implicit deep denoising prior.

**Super-resolution** The noiseless SR problem (Li et al., 2023) consists in recovering $x$ from $y = Ax$, where $A$ is a decimation operator. We underline that this problem can be seen as a special case of inpainting, but where the mask shows a periodic pattern. We fine-tune the meta PDNet model on the same dataset used at the inner level of the MaML task, *i.e.* the 400 natural images from BSD500 and evaluate the test task on the Set3C dataset.

Visual results are provided in Figure 4, and numerical evaluation is reported in Table 1. We notice that the proposed method performs on par with the state-of-the-art (unsupervised) DPIR algorithm. In the unsupervised setting, notice that the model is able to learn an effective model on this unseen task. Training dynamics are available in Figure B.2.

**Magnetic Resonance Imaging** In magnetic resonance imaging (MRI), the sensing device acquires a noisy and undersampled version of the Fourier coefficients of the image of interest. Following the fastMRI approach (Zbontar et al., 2018), we consider a discrete version of the problem where the measurement equation writes $y = MFx$, where $M$ is a binary undersampling mask and $F$ the (discrete) fast Fourier transform. Unlike single-image super-resolution (SR), our model, trained on natural image restoration, faces additional challenges due to substantial distribution shifts not only in the measurement operator, which operates on $\mathbb{C}$, but also in the image domain. The meta PDNet model is fine-tuned on 10 slices from the fastMRI validation dataset, on the same MRI mask that will be used at test time. At test time, the model is evaluated on 10 other slices from the fastMRI validation set, different from those used during training.

Table 1: Reconstruction metrics for different methods on the two problems studied in this paper. PDNet refers to the architecture trained on the testing task, while the MaML (supervised and unsupervised) versions are fine-tuned on the task of interest with only 50 steps of Adam.

|  | wavelets | TGV | DPIR | PDNet-MaML sup., 50 steps | PDNet-MaML unsup., 50 steps | PDNet |
|---|---|---|---|---|---|---|
| SR ×2 | $23.84 \pm 1.22$ | $25.52 \pm 1.33$ | $27.34 \pm 0.67$ | $26.73 \pm 0.78$ | $\mathbf{27.36 \pm 0.79}$ | $28.20 \pm 0.76$ |
| MRI ×4 | $28.71 \pm 2.05$ | $28.69 \pm 2.10$ | $\mathbf{29.12 \pm 2.18}$ | $\underline{29.04 \pm 1.89}$ | $27.64 \pm 1.97$ | $30.04 \pm 2.08$ |

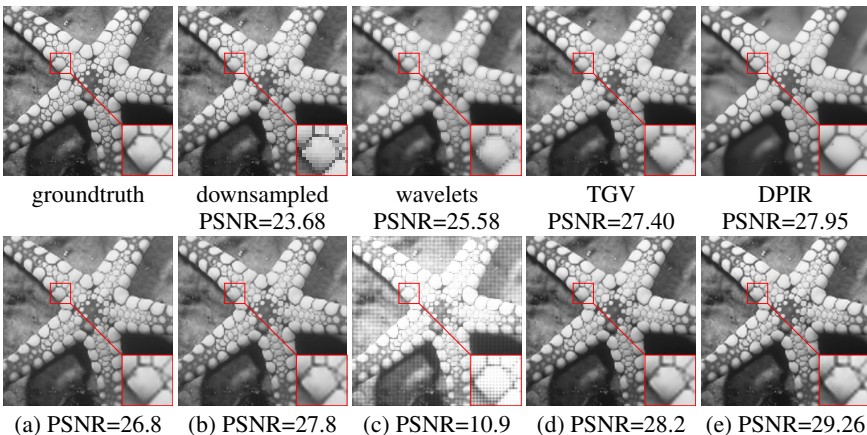

|  |  |  |  |  |
|---|---|---|---|---|
| groundtruth | downsampled PSNR=23.68 | wavelets PSNR=25.58 | TGV PSNR=27.40 | DPIR PSNR=27.95 |
| (a) PSNR=26.8 | (b) PSNR=27.8 | (c) PSNR=10.9 | (d) PSNR=28.2 | (e) PSNR=29.26 |

Figure 4: Results on the SR test set. (a) Result after 1 step of supervised training. (b) Result after 20 steps of supervised training. (c) Result after 1 step of unsupervised training. (d) Result after 20 steps of unsupervised training. (e) PDNet trained with random initialization for 10k steps.

We provide results in Table 1 for simulations with an acceleration factor 4. While the supervised MaML approach performs on par with DPIR, its unsupervised version fails to learn meaningful results. This shows the limit of the proposed method; we suspect that the poor performance of the unsupervised fine-tuning is more a consequence of the distribution shift between the measurement operators than the distribution shift between the sought images. Training dynamics are available in Figure B.2 and visual results are reported in Figure B.3.

## 5 CONCLUSION

In this paper, we have introduced a meta-learning approach designed for solving inverse imaging problems. Our approach harnesses the versatility of meta-learning, enabling the simultaneous leveraging of multiple tasks. In particular, each fine-tuning task benefits from information aggregated across the diverse set of inverse problems encountered during training, yielding models easier to adapt to novel tasks. A specific feature of our approach is that the fine-tuning step can be performed in an unsupervised way, enabling to solve unseen problems without requiring costly ground truth images. Our methodology relies on an unrolled primal-dual network, showcasing the efficiency of the meta-learning paradigm for fine-tuning neural networks. This efficiency holds true even when faced with test tasks that significantly diverge from the training dataset. Yet, while it yields promising results with unsupervised learning in settings akin to the training data distribution, our proposed approach encounters notable challenges when extending its applicability to substantially out-of-distribution problems, as often encountered in the domain of MRI applications.

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

# A   TECHNICAL RESULTS

## A.1   CHARACTERIZING THE META-PRIOR GRADIENTS

**Theorem 3.1.** *Consider Problem* (3) *and assume that for all* $i$, $y_i \in \mathrm{Im}(A_i)$. *Then*

(i) *During fine-tuning on a task* $\mathcal{T}_i$ *(in either supervised or unsupervised settings), the fine-tuned weight* $\theta_i$ *satisfies* $\Pi_{\mathrm{Ker}(A_i)}\theta_i = \Pi_{\mathrm{Ker}(A_i)}\theta^*$.

(ii) *Moreover, if the fine-tuning is performed with gradient descent and initialized at* $\theta^*$, *it holds at any step* $t \in \mathbb{N}$ *during optimization that the iterates* $(\theta_i^{(t)})_{t\in\mathbb{N}}$ *satisfy* $\Pi_{\mathrm{Ker}(A_i)}\theta_i^{(t)} = \Pi_{\mathrm{Ker}(A_i)}\theta^*$.

(iii) *Assume that* $\bigcap_i \mathrm{Ker}(A_i) = \{0\}$ *and that* $\sum_j A_i x_i^{(j)} x_i^{(j)\top} A_i^\top$ *is full rank. Then the outer-loss for training the meta-model admits a unique minimizer.*

*Proof of Theorem 3.1.* (i) We consider the task of fine-tuning the model $f_\theta(y) = \theta y$ on a new test task $\mathcal{T}_J$ in an unsupervised setting starting from the meta-model $\theta^*$. The gradient of the inner loss writes

$$\nabla_\phi g_J(x, \phi) = A_J^\top A_J \phi A_J x x^\top A_J^\top - 2 A_J^\top A_J x x^\top A_J^\top + \phi - \theta^*. \tag{9}$$

Noticing that the two leftmost terms in (9) cancel on $\mathrm{Ker}(A_J)$, we deduce from the optimality condition $\nabla_\phi g_J(x, \theta_J) = 0$ that $\Pi_{\mathrm{Ker}(A_J)}(\theta_J - \theta^*) = 0$, hence the result.

(ii) Furthermore, if the minimization of the fine-tuning loss $g_J$ is performed with gradient descent with stepsize $\gamma$ and initialized at $\theta^*$, the iterates $\left(\theta_J^{(t)}\right)_{t\in\mathbb{N}}$ satisfy

$$\theta_J^{(t+1)} = \theta_J^{(t)} - \gamma A_J^\top A_J \theta_J^{(t)} A_J x x^\top A_J^\top - \gamma 2 A_J^\top A_J x x^\top A_J^\top - \gamma(\theta_J^{(t)} - \theta^*). \tag{10}$$

Projecting on $\mathrm{Ker}(A_J)$ yields

$$\Pi_{\mathrm{Ker}(A_J)}\left(\theta_J^{(t+1)}\right) = \Pi_{\mathrm{Ker}(A_J)}\left((1-\gamma)\theta_J^{(t)} + \gamma\theta^*\right). \tag{11}$$

At initialization, $\theta_J^{(0)} = \theta^*$; we conclude by recursion that for all $t \in \mathbb{N}$,

$$\Pi_{\mathrm{Ker}(A_J)}\left(\theta_J^{(t)}\right) = \Pi_{\mathrm{Ker}(A_J)}\left(\theta^*\right). \tag{12}$$

The same results can be obtained in the supervised setting. In this case, the inner gradient writes

$$\nabla_\phi g_J(x, \phi) = \phi A_J x x^\top A_J^\top - x x^\top A_J^\top + \phi - \theta^*, \tag{13}$$

and the results follow from similar arguments.

(iii) Now, let us denote, for all $1 \le i \le I$, $g_i \colon \phi \mapsto \frac{1}{2}\|y_i - A_i\phi y_i\|_2^2 + \frac{1}{2}\|\phi - \theta\|_2^2$. One has

$$\nabla g_i(\phi) = A_i^\top A_i \phi y_i y_i^\top - 2 A_i^\top y_i y_i^\top + \lambda(\phi - \theta). \tag{14}$$

The optimality condition of $\theta_i$ thus writes

$$A_i^\top A_i \theta_i y_i y_i^\top + \lambda\theta_i = \lambda\theta + 2 A_i^\top y_i y_i^\top \tag{15}$$

This equation is a special form of a linear system (Stein's equation). In our case, we know that such a solution exists since it is the solution to a strongly convex minimization problem. However, the

closed-form solution is difficult to decipher. The problematic term in this equation is $y_i y_i^\top$, which enforces rank restrictions on the solution. Nevertheless, the solution of this linear system is linear in $\theta$ and thus

$$\theta_i = K_i \theta + c_i \ . \tag{16}$$

Note that if we neglect the term $y_i y_i^\top$ in the system of equation –or equivalently, replace it with the identity– we get that $K_i = \left(\lambda \operatorname{Id} + A_i^\top A_i\right)^{-1}$. It is to be noted that these matrices are full rank as soon as $\lambda > 0$. Problem (2) can thus be rewritten as

$$\theta^* = \arg\min_\theta \frac{1}{2I} \sum_{i=1}^I \|\widetilde{x}_i - K_i \theta_i y_i\|_2^2, \tag{17}$$

the optimality condition of which writes

$$\sum_i K_i^\top K_i \theta^* y_i y_i^\top - K_i^\top \widetilde{x}_i y_i^\top = 0. \tag{18}$$

Assume now that $y_i$ belongs to rank of $A_i$, *i.e.* that there exists $x_i$ s.t. $y_i = A_i x_i$; we have

$$\nabla \mathcal{L}_\theta(y, x) = \sum_i K_i^\top K_i \theta A_i x_i x_i^\top A_i^\top - K_i^\top \widetilde{x}_i x_i^\top A_i^\top \tag{19}$$

Assume now that $\bigcap_i \operatorname{Ker}(A_i) = \{0\}$ and, without loss of generality, that $\operatorname{Ker}(A_i) \bigcap \operatorname{Ker}(A_j) = \{0\}$ for $i \neq j$. Assume eventually that for every $i$, $\sum_j A_i x_j^{(i)} x_j^{(i)\top} A_i^\top$ is full rank. Then for every $i$, the optimality condition yields a system of $n \times m_i$ unknowns with $n \times m_i$ equations. Summing over the range of each $A_i$, we obtain a system of $n \times m$ equations with $n \times m$ unknowns the solution of which has a unique solution.

$\square$

## A.2 Linear Inverse Problem's Bayes Predictor

We now give the proof of Lemma 3.2:

**Lemma 3.2.** *Let $A_i$ a linear operator and assume that $x \sim \mathcal{N}(\mu, \Sigma)$ and $y = A_i x$. Then the Bayes' estimator (4) satisfies:*

$$\begin{cases} \widehat{x}(y, A_i)_{\operatorname{Im}(A_i^\top)} = A_i^\dagger y\big|_{\operatorname{Im}(A_i^\top)}, \\ \widehat{x}(y, A_i)_{\operatorname{Ker}(A_i)} = \mu_{\operatorname{Ker}(A_i)} + \Sigma_{\operatorname{Ker}(A_i), \operatorname{Im}(A_i^\top)} \left(\Sigma_{\operatorname{Im}(A_i^\top)}\right)^{-1} (A_i^\dagger y - \mu_{\operatorname{Im}(A_i^\top)}), \end{cases} \tag{5}$$

*where we have used the decomposition*

$$\mu = \begin{pmatrix} \mu_{\operatorname{Im}(A_i^\top)} \\ \mu_{\operatorname{Ker}(A_i)} \end{pmatrix} \quad and \quad \Sigma = \begin{pmatrix} \Sigma_{\operatorname{Im}(A_i^\top)} & \Sigma_{\operatorname{Im}(A_i^\top), \operatorname{Ker}(A_i)} \\ \Sigma_{\operatorname{Ker}(A_i), \operatorname{Im}(A_i^\top)} & \Sigma_{\operatorname{Ker}(A_i)} \end{pmatrix}.$$

*Proof of Lemma 3.2.* Without loss of generality, we can decompose $x \sim \mathcal{N}(\mu, \Sigma)$, $\mu$ and $\Sigma$ on $\operatorname{Im}(A^\top) \oplus \operatorname{Ker}(A)$ as follows:

$$x = \begin{pmatrix} x_{\operatorname{Im}(A_i^\top)} \\ x_{\operatorname{Ker}(A_i)} \end{pmatrix}, \quad \mu = \begin{pmatrix} \mu_{\operatorname{Im}(A_i^\top)} \\ \mu_{\operatorname{Ker}(A_i)} \end{pmatrix}, \quad \Sigma = \begin{pmatrix} \Sigma_{\operatorname{Im}(A_i^\top)} & \Sigma_{\operatorname{Im}(A_i^\top), \operatorname{Ker}(A_i)} \\ \Sigma_{\operatorname{Ker}(A_i), \operatorname{Im}(A_i^\top)} & \Sigma_{\operatorname{Ker}(A_i)} \end{pmatrix} \ . \tag{20}$$

Let us first consider the component on $\operatorname{Im}(A_i^\top)$. As $A_i^\dagger$ is a deterministic function of $y$ which is injective, we have

$$\widehat{\theta}(y, A_i)\big|_{\operatorname{Im}(A_i^\top)} = \mathbb{E}\left[x_{\operatorname{Im}(A_i^\top)}\big|y\right] = \mathbb{E}\left[x_{\operatorname{Im}(A_i^\top)}\big|A_i^\dagger y\right] \ . \tag{21}$$

By definition of the Moore-Penrose pseudo-inverse, we have $x_{\operatorname{Im}(A_i^\top)} = (A_i^\dagger y)\big|_{\operatorname{Im}(A_i^\top)}$, and therefore the first part of the result.

We now turn to the second component of the Bayes' estimator. First, let us consider the conditional expectation of $x_{\text{Ker}(A_i)}$ given $x_{\text{Im}(A_i^\top)}$. Simple computations, similar to the ones from Le Morvan et al. (2020, Proposition 2.1), show that

$$\mathbb{E}[x_{\text{Ker}(A_i)}|x_{\text{Im}(A_i^\top)}] = \mu_{\text{Ker}(A_i)} + \Sigma_{\text{Ker}(A_i),\text{Im}(A_i^\top)}\left(\Sigma_{\text{Im}(A_i^\top)}\right)^{-1}(x_{\text{Im}(A_i^\top)} - \mu_{\text{Im}(A_i^\top)}) \ . \quad (22)$$

Using again that $A_i^\dagger$ is a deterministic and injective function of $y$, we have

$$\widehat{\theta}(y, A_i)\big|_{\text{Ker}(A_i)} = \mathbb{E}\left[x_{\text{Ker}(A_i)}\big|A_i^\dagger y\right] = \mathbb{E}\left[x_{\text{Ker}(A_i)}\big|x_{\text{Im}(A_i^\top)}\right] \ . \quad (23)$$

Replacing $x_{\text{Im}(A_i^\top)}$ with $A_i^\dagger y$ in (22) gives the expression of the Bayes' estimator. □

## A.3 LINK BETWEEN LINEAR AND NON-LINEAR MODELS

**Proposition 3.3.** *Consider Problem* (2). *If the network $f_\theta$ is extremely overparametrized and $\Pi_{\text{Ker}(A_i)}$ commutes with $J_{\theta^*} J_{\theta^*}^\top$, then, we have*

$$f_\theta(y)\big|_{\text{ker}(A_i)} = f_{\theta^*}(y)\big|_{\text{ker}(A_i)} \ ,$$

*when the inner loss is unsupervised.*

*Proof of Proposition 3.3.* The optimality condition of the inner problem implies

$$J_\theta^\top A_i^\top (A_i f_{\theta_i}(y_i) - y_i) + (\theta_i - \theta^*) = 0, \quad (24)$$

which implies

$$A_i^\top (A_i f_{\theta_i}(y_i) - y_i) + \left(J_\theta^\top\right)^\dagger (\theta_i - \theta^*) = 0. \quad (25)$$

Now, assume that $A_i f_{\theta_i}(y_i) \neq y_i$; then the left hand-side belongs to $\text{Im}(A_i^\top) = \text{Ker}(A_i)^\perp$, hence $\left(J_\theta^\top\right)^\dagger (\theta_i - \theta^*)\big|_{\text{Ker}(A_i)} = 0$. Multiplying by $J_\theta J_\theta^\top$ and assuming that $J_\theta J_\theta^\top$ commutes with $\Pi_{\text{Ker}(A_i)}$, we have that $J_\theta J_\theta^\top \left(J_\theta^\top\right)^\dagger (\theta_i - \theta^*)\big|_{\text{Ker}(A_i)} = 0$; eventually, since for any linear operator $L$, one has $L^\top = L^\top L L^\dagger$, we conclude that

$$J_\theta(\theta_i - \theta^*)\big|_{\text{Ker}(A_i)} = 0 \quad (26)$$

A first-order Taylor expansion in $\theta_i = \theta^* + \Delta\theta$ gives

$$f_{\theta_i}(y) = f_\theta(y) + J_\theta \Delta\theta + o(\|\Delta\theta\|) \quad (27)$$

where $J_\theta \Delta\theta \in \text{Ker}(A_i)^\perp$. It can be shown that, in the overparmetrized regime, this linearization holds in good approximation (Chizat & Bach, 2018). Thus, we deduce that

$$f_{\theta_i}(y)\big|_{\text{Ker}(A_i)} = f_{\theta^*}(y)\big|_{\text{Ker}(A_i)}. \quad (28)$$

□

## B FURTHER EXPERIMENTAL RESULTS

We show in Figure B.1 the results of the meta and inner models for an inpainting (training) task $\mathcal{T}_4$ on the Set3C test set. These results are in line with those shown in Figure 3: the meta-model in the fully supervised setting performs less well than the meta-model in the unsupervised setting, which is approximately on par with the inner model. The supervised inner model performs better than its unsupervised counterpart.

Figure B.2 gives the evolution of the reconstruction PSNR as a function of the number of fine-tuning steps and depth of the model. In the supervised setting, we notice a significant improvement in the reconstruction quality after a single fine-tuning step compared to the results obtained with a network initialized at random. This shows that even in the supervised setup, the benefit of the meta-model is

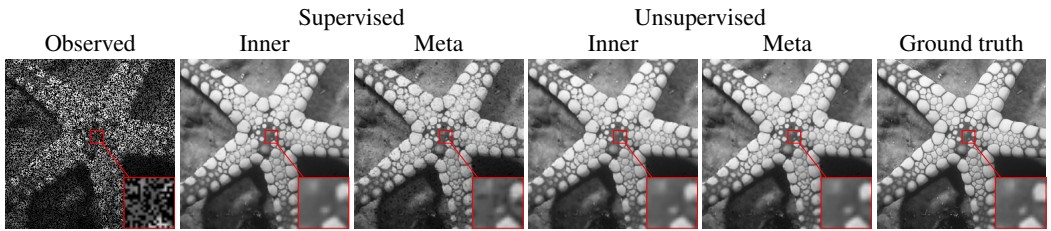

Figure B.1: Results on the validation set on an inpainting operator seen during training.

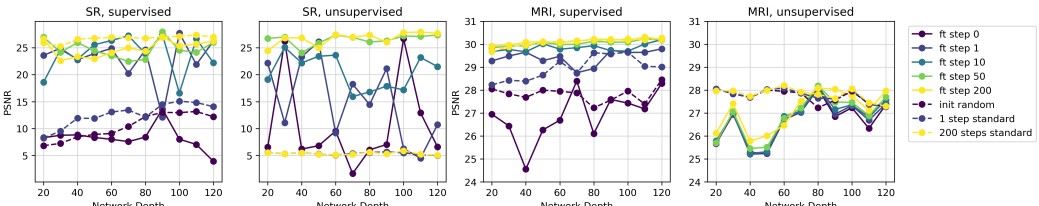

Figure B.2: Results on a ×2 super resolution problem (left) and on an MRI problem (right). On each plot, we display the reconstruction metrics for both the meta model, and models finetuned with different number of steps from the meta model. The supervised (resp. unsupervised) plots refer to networks trained and finetuned with supervised (resp. unsupervised) inner losses.

that it can be adapted to an unseen task quicker than an untrained network, by avoiding retraining a network from scratch. Furthermore, the results of the meta-model show greater volatility but quickly improve as the number of fine-tuning steps increases. We observe a different behaviour in the case of the MRI problem. In the supervised setting, few fine-tuning steps are sufficient to improve the reconstruction quality significantly. Yet, we notice that fine-tuning the model in a purely unsupervised fashion does not yield any noticeable improvement over an untrained network.

Visual results for MRI reconstructions are provided in Figure B.3. We notice that while the meta-models do not provide improvements in PSNR metric over the observed data, it does slightly reduce the blurring in the image. However, after 1 step of fine-tuning, we notice a reduction in the Fourier artifacts present in the reconstruction, which is further improved after 20 steps. We did not witness any significant improvement between the reconstruction from the meta-model and the fine-tuned model in the unsupervised setting. This experiment suggests that the meta-model does provide a trained state $\theta^*$ that contains valuable prior information.

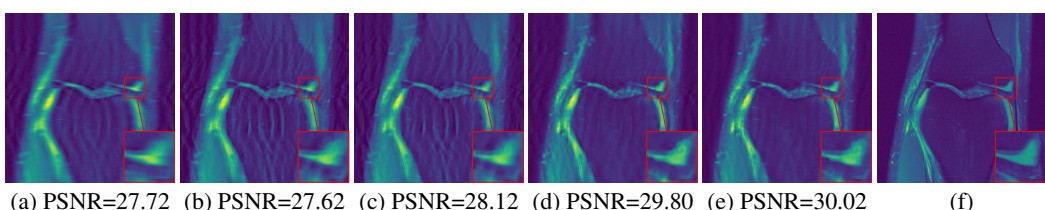

(a) PSNR=27.72  (b) PSNR=27.62  (c) PSNR=28.12  (d) PSNR=29.80  (e) PSNR=30.02  (f)

Figure B.3: Results on a sample from the MRI test set: (a) Simulated MRI scan with a factor 4 speedup. (b) Meta model after unsupervised training. (c) Meta model after supervised training. (d) Results after 1 step of supervised fine-tuning. (e) Results after supervised 20 steps of fine-tuning. (f) Ground truth. PSNR metric is indicated below each image.

