# OpenReview forum: "Meta-Prior: Meta learning for Adaptive Inverse Problem Solvers"
_ICLR.cc/2024/Conference — Submitted to ICLR 2024_

### Official Review · Reviewer_r4M5 · 2023-10-31

**Soundness:** 2 fair
**Presentation:** 4 excellent
**Contribution:** 2 fair
**Rating:** 5
**Confidence:** 4

**Summary:**

This paper proposed a meta-prior model that leverages meta-learning for adaptive inverse problem solvers. The authors explored both supervised and unsupervised settings for the inner loop. The proposed model can be generalized to a new imaging task with as few as 1 step. The experiments were conducted on four diverse imaging tasks and evaluated on image super-resolution and MRI tasks.

**Strengths:**

1. This paper explored the meta learning for imaging tasks, which can be quickly adapted to a new imaging task.
2. The proposed method is supported by theoretical analysis.
3. The experiments cover diverse imaging task.

**Weaknesses:**

1. The experimental results were not promising.
2. Some strong baselines could be considered for comparison.

**Questions:**

1. In Fig. 2, the similarity between learned and analytic solution seems not close. Absolute difference map may be helpful to visualize the difference.
2. In Fig. 3, the proposed meta-model performs worse than task-specific models.
3. For the generalization experiments, (i) quantitative results were missing, (ii) results in Fig. 6 were not promising as there is clear shift from ground-truth, and (iii) I think some unsupervised or self-supervised baselines could be compared as there should be more data for unsupervised training in imaging tasks.

---

> ### Author Response · Authors · 2023-11-20
>
> We thank the reviewer for taking the time to review our work and for their constructive feedback. Please find our response below:
>
> **W1. Unpromising experimental results.** We respectfully disagree with the reviewer. Our experiments show that the meta-learning framework can be successfully applied to the inverse imaging problem, and that it can even be leveraged to effectively learn to solve inverse imaging problems in a fully unsupervised setting.
>
> **Q1. Difference between learned and analytic solution.** We appreciate the reviewer's keen observation regarding the dissimilarity between the learned and analytic solutions in Fig. 2. The noted differences are indeed present, and we acknowledge the stochastic nature of the task as a contributing factor. Unfortunately, due to space constraints, we are unable to include an additional figure displaying the absolute difference map. However, we have added a mention to this observation in the text, emphasizing the inherent stochasticity of the task and its impact on the learned solution.
>
> **Q2. Worse performance of meta-model.** In Figure 3, we evaluate the performance of the meta-model on each finetuning task. Note that the meta-model is the model with “barycenter weights” that is meant to perform averagely well on the sum tasks (this corresponds to the outer loss function). As a consequence, it is expected that the inner model (blue curve) performs worst than the meta model (orange curve).
>
> **W2 & Q3. Comparison with additional methods.** We agree with the reviewer that the submitted paper was missing comparisons. We have added a comparison with several methods, namely standard regularization-based approaches, which are widely used in real imaging settings due to their versatility, as well as with the DPIR method, which reaches state-of-the-art performance in synthetic imaging problems. Note that these methods can be seen as unsupervised techniques. up to a single scalar regularization parameter requiring to be fine-tuned. The proposed method performs on par with these different algorithms. Furthermore, we also compare the results obtained by our backbone architecture when trained in a classical supervised fashion on the test task. As expected, this latter approach outperforms the proposed method.

---

### Official Review · Reviewer_JGWM · 2023-11-01

**Soundness:** 2 fair
**Presentation:** 3 good
**Contribution:** 2 fair
**Rating:** 3
**Confidence:** 3

**Summary:**

The paper proposes a meta model training procedure, that trains a deep models simultaneously on different inverse problems (denoting, restoration with total variation regularization, deconvolution, and image inpainting) while making sure the fine-tuned models (Starting from the meta-model) performing well. The paper formulates the problem first in general form, then it considers some simplified cases where it approaches the optimal bayesian rule. The paper evaluates the performance on the super resolution and MRI reconstruction task.

**Strengths:**

The paper presents a novel approach to meta-model training that is worth considering. It is well-written and technically sound.

One notable strength of the paper is its ability to demonstrate the method's effectiveness in simplified cases where it converges to the optimal estimator. This simple illustration is important as it underscores the method's potential utility in real-world applications.

The paper also offers insightful perspectives, particularly in discussing the relationship between the solution and the kernel space. This insight serves as a motivation for the approach.

**Weaknesses:**

One significant point of criticism in the paper relates to its evaluation process, which has certain shortcomings.

Firstly, the paper lacks clarity in specifying the specific datasets used for evaluating the method's generalization capabilities. Vital details (such as the dataset sizes) are missing. This omission makes it challenging for readers to gauge how the proposed method performs in different real-world scenarios or how representative the result is.

Another notable issue is the limited comparative analysis. The paper primarily focuses on comparisons among different versions of its own method (with varying numbers of fine-tuning steps) and random initialization. However, the absence of comparisons to basic pre-training methods [e.g. as suggested in Chapter 8.7.1 of "Deep Learning" by Goodfellow et al. (2016)] narrows the paper's significance. A more comprehensive evaluation, including comparisons to established pre-training techniques, would offer a better understanding of the proposed method's effectiveness and its relevance within the broader machine learning field.

**Questions:**

n/a

---

> ### Author Response · Authors · 2023-11-20
>
> We thank the reviewer for taking the time to read our work and for their constructive feedback. Please find our response below.
>
> **1. Precisions on datasets & baselines.** We thank the reviewer for raising this point, which is in line with the comment of reviewer jubY. We have added the missing information regarding the datasets to the paper. Our experiments in the super-resolution setting confirm that the MAML framework can be leveraged to learn image reconstruction networks while enforcing data consistency only at fine-tuning time. However, we do not make the same observation on the MRI problem, which may be explained by the strong distribution shift between the training and testing tasks and datasets.
>
> **2. Comparison with additional methods.** We agree with the reviewer that the submitted paper was missing comparisons. We have added a comparison with several methods, namely standard regularization-based approaches, which are widely used in real imaging settings due to their versatility, as well as with the DPIR method, which reaches state-of-the-art performance in synthetic imaging problems. Note that these methods can be seen as unsupervised techniques, up to a single scalar regularization parameter requiring to be fine-tuned. The proposed method performs on par with these different algorithms. Furthermore, we also compare the results obtained by our backbone architecture when trained in a classical supervised fashion on the test task. As expected, this latter approach outperforms the proposed method.

---

### Official Review · Reviewer_jubY · 2023-11-01

**Soundness:** 2 fair
**Presentation:** 3 good
**Contribution:** 2 fair
**Rating:** 3
**Confidence:** 4

**Summary:**

This paper presents a meta-learning approach to solve inverse problems, considering both supervised or unsupervised inner-optimization in a MAML-based meta-formulation. The authors examined the theoretical properties of this “meta-prior”, and experimentally examined its effectiveness compared to learning from scratch in a collection of imaging tasks (de-nosing, TV recovery, deconvolution, inpainting, SR, and MRI reconstruction).

**Strengths:**

The idea of using a meta-learning approach to learn to solve inverse problems across a set of tasks is interesting.

The relatively high diversity of imaging tasks considered in the experimentation is appreciated.

**Weaknesses:**

Two main feedback about the contribution of this week, despite the interesting idea, is the limited methodological novelty and the limited experimental evaluations.

1. The proposed method is primarily a direct application of MAML to image reconstruction tasks — what are the potential challenges in this application and what novel solutions are needed to overcome these challenges are not clear. See some of the questions in the next block.

2. The experimental evaluation is very limited and the descriptions lack many details (see detailed questions below). Furthermore, even just compared to training from scratch on test data only (which is a weak baseline), the benefit of the proposed meta-approach is not significant nor consistent.  Please see the detailed questions below.

**Questions:**

1. Questions regarding methodology.
a. What are the restrictions on the size of A? Does it have to be the same? How are these achieved when x’s and y’s across different image reconstruction tasks are of different dimensions? More generally, what are the requirements on A across tasks?
b.  The stability and cost of MAML training, especially regarding which portion of the primary model to update during inner optimization, is a non-trivial issue. In PDNet, which part of the architecture is being fine-tuned in the inner optimization?

2. Questions regarding experimentations.
a. It is not clear what are the number of data samples used on each training task; what is the size of the context data (referred to as training data in the paper) for each task versus the query data (referred to as test data in the paper). What about the fine-tuning at test time — what are the number of data used in training/fine-tuning, and how many samples are used for evaluating testing performance, throughout all experiments including tasks seen in training and those unseen.
b. On tasks included in training, fine-tuning from meta-models seems to be only compared to the meta-model itself. To understand the contribution of the work, it will be necessary to compare the fine-tuned models to 1) task-specific models trained on the same meta-training data, and 2) models that are trained on the overall meta-training data across all tasks, without and with fine-tuning to the test data used in each tasks.
c. On tasks included in testing, fine-tuned models are only compared to training from scratch on the test data. Again, baselines are needed that considered the same meta-training data (as listed above).
d. Please clarify, when test-time fine-tuning is unsupervised, does that mean the meta-training also considers unsupervised fine-tuning in the inner optimization?
d. The benefits of the presented model needs to be better highlighted.

---

> ### Author Response · Authors · 2023-11-20
>
> We thank the reviewer for taking the time to read our work and for providing constructive feedback. Please find our response below.
>
> **1. Methodology**
>
> **1.a Restrictions on the measurement operator A.** Our current implementation has only been trained on image-to-image tasks of the same shape; however, this could be waived in a future implementation. Moreover, there is however no restriction on the dimension and nature of the data used at the fine-tuning stage due to the versatile backbone primal-dual network. For instance, in the MRI experiment, the data of interest lives in $\mathbb{C}$ while the network was trained on variables in $\mathbb{R}$. So there are no direct requirements on $A$ to apply our framework. However, as exposed in Theorem 3.1, the larger the kernel of $A$, the more diverse the task needs to be for the model to learn to complete the missing information and give good performances.
>
> **1.b MAML training cost.** The full model is being fine-tuned at the level of the inner optimization. We acknowledge that MAML is a heavy training procedure; however, PDNet is a light model with only 361 trainable parameters per layer, i.e. a maximum of 38k parameters in its deepest configuration. We however agree that for larger models, such as UNets, the approach may face memory issues, which were not a problem in the current setting. We added a comment on this point, thanks for pointing it out.
>
> **2. Experiments**
>
> **2.a Clarification on the training data.** We thank the reviewer for pointing this out; we have amended the paper to clarify this matter (see “Training data” as well as experiments’ specific paragraphs). Our network is trained on the BSD500 training and test set combined, containing 400 natural images, and on which all of the 4 inner problems are solved; the outer loss is computed on the test set of the BSD500 images, containing 100 natural images. Finetuning: (i) for SR the model is finetuned on the same dataset used at the inner-level of the MAML task, i.e the 400 natural images from BSD500. The test task is evaluated on the Set3C dataset. (ii) for MRI the model is finetuned on 10 slices from the fastMRI validation dataset, on the same MRI mask that will be used at test time. At test time, the model is evaluated on (different) 10 slices from the fastMRI validation set.
>
> **2.b/c Comparison with the task-specific model.** We thank the reviewer for the suggestion; we have added in Table 1 a comparison with the PDNet architecture trained on the training set (inner level problem) for the test task of the BSD400 dataset for the SR task, and on the fine-tuning set of the MRI problem. This provides a fair upper bound on the best performance achievable by a model trained specifically on the task of interest. Our experiments show that the MAML model does not perform as well, but performs on par with other baseline unsupervised approaches that were also added to the paper.
>
> **2.d Unsupervised fine-tuning** Yes, in the unsupervised setting, the inner optimization is also unsupervised. This is stated in the paragraph after equation (2) in the paper, stating:
> ``It can be either the supervised loss $\mathcal{L}\_{\text{sup}}$ or the unsupervised loss $\mathcal{L}\_{\text{uns}}$, with an extra regularization term controlled by $\lambda>0$ to ensure that the model $f_{\theta_i}$ is close to the meta-model $f_{\theta^*}$. When $\mathcal{L}\_{\text{inner}}$ uses $\mathcal{L}\_{\text{sup}}$ (resp. $\mathcal{L}\_{\text{uns}}$), we call this problem the *supervised meta-learning* (resp. *unsupervised meta-learning*). Finally, the outer training loss $\mathcal{L}\_{\text{outer}}$ is used to evaluate how well the model $f_{\theta_i}$ generalizes on the task $\mathcal{T}\_i$, using $\mathcal{L}\_{\text{sup}}$.”
>
> **2.e Benefits of the proposed approach** We believe that the main advantage of our approach lies not so much in the model itself, which is small size and applied to simple imaging tasks, but rather in enabling us to train a model in a situation where no ground truth is available for the specific task considered. A notable instance of this problem is MRI, where the physical acquisition procedure prevents direct access to true data $x$. To learn in such a setting, we leverage tasks for which pairs $(x, y)$ for various sensing operators $A$ are available to propose a principled meta-learning method. Our approach addresses the limitations of handcrafted and implicit priors, and we show theoretically that this framework allows us to learn a prior that is used to complete the missing information in $\operatorname{Ker}(A)$. We have restructured the introduction of the paper to better highlight this motivation.

---

### Official Review · Reviewer_1w5o · 2023-11-02

**Soundness:** 2 fair
**Presentation:** 2 fair
**Contribution:** 2 fair
**Rating:** 3
**Confidence:** 4

**Summary:**

This paper proposes a method of meta learning to train a meta-model on various imaging tasks. The paper claims that finetuning the meta-model makes it easier to perform on new tasks that are unseen during the training.

**Strengths:**

- The proposed method to learn some priors through the meta-model which is expected to generalize across different tasks seems to be interesting.

**Weaknesses:**

- The paper structure and writing can be further improved, especially abstract and introduction. It is confusing to understand what is the motivation and what is novelty even after reading through the introduction.

- Motivation for learning meta-model. Although it looks interesting from the toy model study, why do we want to learn such a meta-model from different tasks? To converge fast or achieve better results in a new task? But why can the meta prior help to achieve this goal? What is the physical meaning of the learned prior, especially when there are large domain gaps between training and testing tasks such as generalized to MRI imaging? What kind of meta training tasks should be helpful and how to choose such tasks? It is not clear about the grounding support for this proposed method.

- No comparison with baseline methods. The proposed method looks like using a very standard meta-learning framework to train this meta-model. What is the novelty and superiority of the proposed method compared to other meta-learning method such as MAML?

- No comparison with other methods in all the application tasks. As shown in Figure 4 and 6, the results demonstration are mainly the analysis within the proposed method with different finetuning steps or training settings. But there is no comparison with other SR methods or MR reconstruction methods. How can it be argued the proposed method can be a good approach to achieve good results on those tasks? For example, as shown in Figure6, the MRI reconstruction results with acceleration factor = 4 seems to be quite lower-quality than many existing methods either supervised or unsupervised.

- I would suggest using a table instead of a figure with so many curves as Figure 5 for results comparison. It is hard to distinguish when many curves are plotted with many overlaps. Showing the network depth may be good additional experiments, but it seems hard to draw some consistent conclusion from these curves.

- Providing a framework figure may be helpful to better illustrate the proposed method especially with meta-training and inner-training.

**Questions:**

Please see weakness for the details of questions.

---

> ### Author Response · Authors · 2023-11-20
>
> We thank the reviewer for taking the time to read our work and providing constructive feedback. Please find our response below.
>
>
> **Unclear motivation.** We have amended the abstract and the introduction to clarify the paper’s goal and the main messages. The main motivation of our paper is learning in contexts where no ground truth is available for a given reconstruction task. This context typically arises in MRI, where only partial measurements for a given k-space sampling scheme $A$ are available. When changing the sampling scheme $A$, one does not have access to the true data $x$ but only to the measurement $y$ and the sensing operator $A$. Thus supervised approaches cannot be used to directly train a network in this setting [1]. However, in many cases one has access to pairs $(x, y)$ for different sensing operators $A$. The motivation of our paper is to propose a principled method to leverage these extra data points. Existing methods either rely on handcrafted priors, that have limited performances, or deep implicit prior, that struggle to generalize to non-toy imaging settings. We propose to frame the problem using meta learning. We show theoretically that this framework allows to learn a prior that is used to complete missing information in $\operatorname{Ker}(A)$ and show that this framework allows to give satisfactory reconstruction. Our motivations are thus three-fold: designing a principled framework for explicit prior learning, pulling information from multiple sources (tasks) and accelerating sensing operator adaptation.
>
>
> **No baseline methods and novelty with regards to MAML.** We have added comparisons with baseline methods for image reconstruction (*see common answer*). The proposed method is an extension of the MAML approach. The novelty in our work is to show how this framework can be adapted in this context, with two main settings (supervised and unsupervised). We further give insights on what this model can learn, and show with a wide variety of experiments that the proposed approach works in practical cases.
>
>
> **Comparison with other methods.** We agree that the initial submission was missing comparisons, which have now been added to the paper. In essence, the proposed method performs on par with the SOTA unsupervised image restoration DPIR algorithm (Zhang et al, 2021) after only 50 steps of finetuning on the test task. However, the proposed model performs less well than its counterpart trained in a supervised setting specifically on the task of interest.
>
>
> **Clarification of Figure 5.** Following the reviewer’s suggestion, we have replaced Figure 5 with a table (see Table 1) including a comparison with various baselines. We believe that this change improves the clarity of the paper.
>
>
> **References**
>
> [1] Shimron, Efrat, et al. "Implicit data crimes: Machine learning bias arising from misuse of public data." Proceedings of the National Academy of Sciences 119.13 (2022)

---

> > ### Comment · Reviewer_1w5o · 2023-11-23
> >
> > Thanks to the author to answer my questions. After reading the response as well as other reviewer's comments, I will keep my original score. Thanks.

---

### Author Response · Authors · 2023-11-20
**General response**

We thank the reviewers for their insightful comments. Our proposed framework is deemed interesting by all reviewers, who appreciated our theoretical insights and the diverse set of empirical validations we provide. The two main criticisms that were raised are (i) some missing details and baselines in the experimental validation and (ii) the lack of clarity regarding the motivations for applying the meta-learning framework to image reconstruction problems. We have addressed these points by (i) adding comparisons with different baselines in the experimental section, and (ii) rewriting some paragraphs regarding our motivations that had raised questions from the reviewers. In a nutshell, the main motivation of our work is to leverage the meta-learning framework to fine-tune a model on an unseen task using measurement consistency only, hence without needing any ground truth data. This use case typically arises when changing the acquisition scheme in MRI. The comparative experiments that were added to the revised version of the paper show that the proposed approach performs comparably to state-of-the-art unsupervised methods. Detailed responses to each of the reviewers’ comments can be found below.

---

### Meta-Review · Area_Chair_9SNz · 2023-12-16

**Metareview:**

The paper proposes to use meta learning to train a meta model on multiple imaging tasks such that fine-tuning the meta-model makes it easier to perform on new tasks. The meta learning problem is posed as a bilevel optimization problem.

Strenghts:
+ The idea of using meta-learning for imaging tasks is interesting

Weaknesses:
- Writing of the paper needs improvement.
- The paper did not provide sufficient details about experiment setup
- No comparison with baseline methods.
- Parts of the paper feel disconnected (e.g., how does the analysis of linear models and linear estimation relate to the experiments)

**Justification For Why Not Higher Score:**

- Disconnect between linear estimation theory and main experiments in the paper.
- Details about experiments and comparison with other methods could have strengthened the paper
- The experiments show incremental improvement of task-specific training. This effect should be studied further and any savings in training time or data should be highlighted
- Some experiments on large-scale imaging problems could be helpful

**Justification For Why Not Lower Score:**

N/A

---

### Decision · Program_Chairs · 2024-01-16

Reject